# Psychometric and biomedical outcomes of glycated haemoglobin target-setting in adults with type 1 and type 2 diabetes: Protocol for a mixed-methods parallel-group randomised feasibility study

**Samuel J. Westall**[1,2]*, **Simon Watmough**[1], **Ram Prakash Narayanan**[2], **Greg Irving**[1], **Kevin Hardy**[2]

1 Faculty of Health, Social Care and Medicine, Edge Hill University, Ormskirk, United Kingdom,
2 Department of Diabetes and Endocrinology, St Helens and Knowsley Teaching Hospitals NHS Trust, St Helens Hospital, St Helens, United Kingdom

* 23723122@edgehill.ac.uk

## Abstract

### Background

The disease burden of diabetes can have wide-ranging implications on patients' psychological well-being and health-related quality of life. Glycated haemoglobin targets are commonly used to guide patient management in diabetes to reduce the future risk of developing diabetes complications, but little is known of the psychological impact of glycated haemoglobin target-setting. This protocol describes a study to determine the feasibility of evaluating psychological outcomes when setting explicit glycated haemoglobin targets in people with diabetes.

### Methods

This single-centre randomised feasibility study will follow a mixed-methods approach across four sub-studies. In sub-study A, eligible adults (aged 18 and over) with type 1 or type 2 diabetes will complete baseline validated psychometric questionnaires evaluating health-related quality of life (EuroQoL-5D-5L), diabetes-related distress (Problem Areas In Diabetes), self-care (Summary of Diabetes Self-Care Activities), well-being (Well-Being Quetionnaire-12) and diabetes-related psychosocial self-efficacy (Diabetes Empowerment Scale-Long Form). Participants will be randomised to receive explicit glycated haemoglobin intervention targets 5mmol/mol above or below current glycated haemoglobin readings. Rates of eligibility, recruitment, retention and questionnaire response rate will be measured. Psychometric outcomes will be re-evaluated 3-months post-intervention. Sub-studies B and C will use qualitative semi-structured interviews to evaluate experiences, views and opinions of diabetes patients and healthcare professionals in relation to the acceptability of study processes, the use of glycated haemoglobin targets, the impact of diabetes on psychological well-being and, in sub-study D, barriers to participation in diabetes research.

**Data Availability Statement:** No datasets were generated or analysed during the current study. All relevant data from this study will be made available upon study completion. All relevant data from this study will be made available on study completion. Anonymised data will be deposited in the Edge Hill University online research data repository, https://figshare.edgehill.ac.uk.

**Funding:** The authors received no specific funding for this work.

**Competing interests:** The authors have declared that no competing interests exist.

## Discussion

This mixed-methods study aims to provide a novel insight into the psychological implications of glycated haemoglobin target-setting for people with diabetes in secondary care, alongside testing the feasibility of undertaking a larger project of this nature.

## Trial registration

The study is registered with the ISRCTN (registration number: 12461724; date registered: 11th June 2021). Protocol version: 2.0.5, 26th February 2021.

## Introduction

### Background

Glycated haemoglobin values are commonly used to guide the management of people with diabetes. Far from being a standardised approach, glycated haemoglobin targets are often individualised to the patient in response to the presence of established cardiovascular disease [1–3], advanced age [4, 5], long diabetes duration [6], frailty [7, 8], excessive comorbidity [9], history of severe hypoglycaemia [10] and comorbid mental health illness [11, 12].

For diabetes healthcare professionals, the importance of a personalised approach to management and goal-setting in people with diabetes has never been clearer, evidenced by the release of the joint American Diabetes Association (ADA)/European Association for the Study of Diabetes (EASD) consensus statement [13, 14]. Over time, glycated haemoglobin is a predictor of risk of developing diabetes complications [15]. Chronically high glycated haemoglobin readings have been shown to negatively impact upon the physical health of people with diabetes: diabetes is one of the leading causes of adult blindness in the UK [16], increases the risk of cancer [17, 18] and dementia [19, 20], more than doubles the risk of heart attack [21], trebles the risk of stroke [21] and is the commonest cause of end-stage renal failure [21] and non-traumatic lower limb amputation in Europe [21].

Several factors have a bearing on the achievement of glycated haemoglobin targets in people with diabetes [22]. Diabetes is recognised to have a significant impact on psychological outcomes and mental health. Indeed, mental health illnesses are seen with increased prevalence in those with diabetes [23]. Prior research has shown that a significant proportion of people with diabetes have depression at a level that impairs activities of daily living, quality of life, adherence to medical treatment, glycaemic control, and increases healthcare utilisation, healthcare cost and the risk of diabetes complications [11, 12, 24, 25]. From a patient's perspective, lower levels of motivation and knowledge and higher levels of mental health comorbidity can create barriers to achieving optimal glycated haemoglobin levels [26]. From a healthcare perspective, contradictory guidelines and complex management options can cause confusion for clinicians in personalising glycated haemoglobin targets in patients [22].

Although awareness of the importance of achieving glycated haemoglobin targets amongst patients with diabetes is improving with patient education [27], many patients remain unaware of the preventable excess risk posed by sub-optimal glycated haemoglobin levels. Despite many new pharmacological interventions that have become available in the past decade, achievement of glycated haemoglobin treatment targets in England and Wales has remained stubbornly below 30 percent for Type 1 Diabetes and below 70 percent for Type 2 Diabetes [28]. Evidence for lack of progress in the achievement of optimal glycated

haemoglobin levels is reflected in studies globally [26]. People with diabetes have historically struggled to access evidenced-based diabetes education (though this is improving) and negative patient perceptions on the intensification of diabetes treatment remain prevalent [26, 29, 30].

Awareness amongst healthcare professionals of the importance using treatment targets to drive improvement in micro- and macro-vascular outcomes is good. Compelling evidence on the use of personalised glycated haemoglobin treatment targets in response to patient factors is gaining traction amongst clinicians in primary and secondary care in the UK. Current literature points to the clear reduction in the risk of diabetes complications by having individualised treatment targets [3, 31–34]. A lack of evidence persists on the impact clinician-initiated glycated haemoglobin target-setting has on psychological and self-management abilities of people with diabetes. It would therefore be valuable to determine the acceptability and preliminary impact of glycated haemoglobin target-setting on the psychological well-being of people with diabetes.

## Study aims

This protocol outlines a feasibility study to achieve the following objectives:

1. Test the feasibility of the study according to eligibility, recruitment and retention.

2. Test the feasibility of data collection procedures.

3. Evaluate the preliminary impact of explicit glycated haemoglobin target-setting (intervention) on health-related quality of life, diabetes-distress, self-care, well-being and diabetes-related psychosocial self-efficacy at 3-month follow-up.

4. Evaluate the acceptability of the intervention and study processes.

5. To understand the experiences, views and opinions of people with diabetes and their healthcare professionals on the acceptability of the study, the intervention and barriers to participation in diabetes research.

## Materials and methods

### Study design

This is a single-centre mixed-method study consisting of an unblinded randomised two-arm feasibility study (sub-study A) and semi-structured interviews (sub-studies B, C and D). A SPIRIT schedule of procedures and events is demonstrated in Fig 1. The 2013 SPIRIT checklist of recommendations [35] is presented in the S1 Table.

### Setting

Recruitment and data collection will be from a secondary care outpatient diabetes centre in the Metropolitan Borough of St Helens, UK. St Helens is in the Northwest UK and comprises of a mixture of rural and urban wards with comparable demographics to other regions in the UK. Recorded semi-structured interviews in the study will take place via telephone.

### Eligibility criteria

For sub-study A, eligible patients will be identified when registered to attend the local secondary care adult diabetes clinic. Adults aged 18 years and over with either Type 1 or Type 2

| Study period | Pre-baseline | Baseline | Intervention | Endpoint | Interviews |
|---|---|---|---|---|---|
| Time point | - | 0 months | 3 months | 6 months | 0–12 months |
| **Screening** | | | | | |
| Screen electronic records | X | | | | |
| Postal invite | X | | | | |
| **Enrolment** | | | | | |
| Check eligibility | | X | | | |
| Informed consent | | X | | | |
| Allocation | | X | | | |
| 3-month run-in | | X ------------------ X | | | |
| **Intervention** | | | | | |
| Group A or B intervention | | | X | | |
| **Assessments** | | | | | |
| POC glycated haemoglobin | | X | X | X | |
| BP, BMI | | X | X | X | |
| EQ-5D-5L, PAID, SDSCA, W-BQ12, DES-LF | | | X | X | |
| Acceptability survey | | | | X | |
| **Additional elements** | | | | | |
| Interview with participants[†] | | | | | X |
| Interview with HCP's[‡] | | | | | X |
| Barriers to participation[§] | | | | | X |

**Fig 1. Schedule of procedures and assessments.** POC: point-of-care; HbA1c: glycated haemoglobin; BP: blood pressure; BMI: body mass index; EQ-5D-5L: EuroQoL-5D-5L; PAID: Problem Areas in Diabetes; SDSCA: Summary of Diabetes Self-care Activities; W-BQ12: Well-being Questionnaire-12; DES-LF: Diabetes Empowerment Scale-Long Form; HCP: Healthcare Professional; mmol/mol: millimoles per mole, glycated haemoglobin SI unit of measurement. † Sub-study B. Semi-structured interviews in those enrolled in sub-study A. ‡ Sub-study C. Semi-structured interviews with healthcare professionals. § Sub-study D. In those declining to take part in sub-study A, survey and semi-structured interview.

diabetes and with a glycated haemoglobin reading at recruitment between 64 and 125 mmol/mol will be eligible for recruitment. Potential participants will be excluded if they: are at risk of cardiovascular disease events; have had an episode of severe hypoglycaemia within the past 12 months; have hypoglycaemia unawareness (defined as Gold score $\geq 4$ [36]); are unwilling to self-monitor blood glucose (if clinical management requires); are unwilling to inject insulin (if clinical management requires); have a body mass index $\geq 45$ kg/m$^2$; have opted-out from being contacted by researchers under the national NHS date opt-out service; have another serious illness which may limit survival; have factors that may limit adherence to study interventions; are currently participating in another trial; are pregnant; have requirement for regular venesection or blood transfusion; or, are undergoing medical therapies known to cause difficulties with glycaemic control (such as corticosteroid therapy). Eligible individuals will be invited to take part in the study. Reasons for decline will be recorded.

A purposive sample of participants enrolled will be invited into sub-study B for interview (see Allocation). Eligible patients declining to take part will be invited into sub-study D, a survey and interview evaluating barriers to participation in diabetes research. A convenience sample of diabetes healthcare professionals will be invited into sub-study C for interview.

## Intervention

**Glycated haemoglobin target-setting.** Explicit glycaemic targets will be given to participants as described using the TIDieR checklist [37] in Table 1.

In primary care in the UK, glycated haemoglobin targets are incentivised under the Quality and Outcomes framework. Recruitment of participants is undertaken from secondary care, therefore primary care target incentivisation is not anticipated to impact the intervention

**Table 1. Study intervention described using the TIDieR checklist.**

| Item | Description |
|---|---|
| **1. Name of intervention** | Group A: In-person (1-on-1) reset of glycaemic target to 5mmol/mol above current participant glycated haemoglobin reading. |
| | Group B: In-person (1-on-1) reset of glycaemic target to 5mmol/mol below current participant glycated haemoglobin reading. |
| **2. Rationale** | The psychological effect of glycaemic target-setting in people with diabetes is unknown. |
| **3. Materials** | A supplemental study leaflet shows a visual representation of participants' current HbA1c result (on a scale from 'non-diabetic' to 'very high glycated haemoglobin levels') alongside participants' study target glycated haemoglobin. |
| **4. Procedures** | The intervention takes place as a discussion between the intervention provider and the participant. The discussion is guided by pre-defined script. |
| **5. Intervention provider** | The intervention is provided by the Main Investigator. |
| **6. Mode of intervention delivery** | The intervention is delivered face-to-face on an individual basis. |
| **7. Location** | The intervention is delivered in a private consultation room within a secondary care diabetes centre in the Northwest UK. |
| **8. Time requirements/timing** | The intervention takes place 3-months post-allocation to allow for initial secondary care clinical management (undertaken at study entry) to be fully established. Intervention delivery is expected to take 10–15 minutes as a discussion between researcher and participant. |
| **9. Personalisation** | Not applicable. |
| **11. Ensuring intervention adherence** | A single intervention provider delivers the intervention in order to ensure rigorous and consistent adherence to the protocol for the duration of the trial. |

Items 10 and 12 are excluded as they are not applicable until the study is complete.

during the study period. As shown in Table 1, a supplemental glycated haemoglobin fillable leaflet has been developed to visually represent the new glycated haemoglobin target study intervention to participants. The leaflet will be used in conjunction with a face-to-face intervention delivery; setting participants' glycaemic haemoglobin target 5 mmol/mol above (in those randomised to Group A) or below (Group B) current glycated haemoglobin readings (worked example, Table 2). Intervention-related new glycated haemoglobin levels will be fixed for the remainder of the study after which they will be returned to pre-study levels.

**Treatment as usual.** With the primary aim of this study testing feasibility—'can this trial be done?'—a control group will not be used. Participants' usual diabetes care will continue. Any participants highlighted by the outcome tools as needing psychological support will be signposted to relevant local providers.

## Recruitment and follow-up

The schedule of procedures and events is presented in Fig 1, in accordance with the 2013 SPIRIT Statement [35]. A flowchart of study processes is demonstrated in Fig 2. Eligible participants will be invited via post to sub-study A following review of upcoming secondary care diabetes patient clinic lists. Following diabetes clinic appointments, invited persons will be screened to confirm eligibility. Written, informed consent will be obtained from eligible individuals willing to consent to study participation. Following consent, participants will be randomly allocated to group A or B (see Allocation). Trained clinic staff (registered nurses and healthcare assistants) obtain point-of-care glycated haemoglobin readings, blood pressure and body mass index for the study at enrolment, intervention and follow-up.

**Table 2. Worked example of sub-study A intervention.**

| Study period | Pre-baseline | Baseline | Intervention | Endpoint |
|---|---|---|---|---|
| Timepoint | - | 0 months | 3 months | 6 months |
| **Intervention Group A** | | | | |
| Example patient 'A' glycated haemoglobin level | 65 mmol/mol | 67 mmol/mol | 63 mmol/mol | 62 mmol/mol |
| Glycated haemoglobin target | < 58 mmol/mol | < 58 mmol/mol | ↑ < 68 mmol/mol[a] | < 58 mmol/mol |
| **Intervention Group B** | | | | |
| Example patient 'B' glycated haemoglobin level | 77 mmol/mol | 78 mmol/mol | 74 mmol/mol | 65 mmol/mol |
| Glycated haemoglobin target | < 53 mmol/mol | < 53 mmol/mol | ↓ < 69 mmol/mol[b] | < 53 mmol/mol |

[a] Group A study intervention: glycated haemoglobin target increased by 5 mmol/mol against patient's current glycated haemoglobin level.

[b] Group B study intervention: glycated haemoglobin target decreased by 5 mmol/mol against patient's current glycated haemoglobin level.

A 3-month run-in period is allowed to optimise participant clinical management prior to intervention delivery. Following the run-in period, participants will complete baseline psychometric questionnaires after which the study intervention will be delivered according to allocation group.

At the study endpoint, 3-months post-intervention, participants complete psychometric questionnaires and a study acceptability survey.

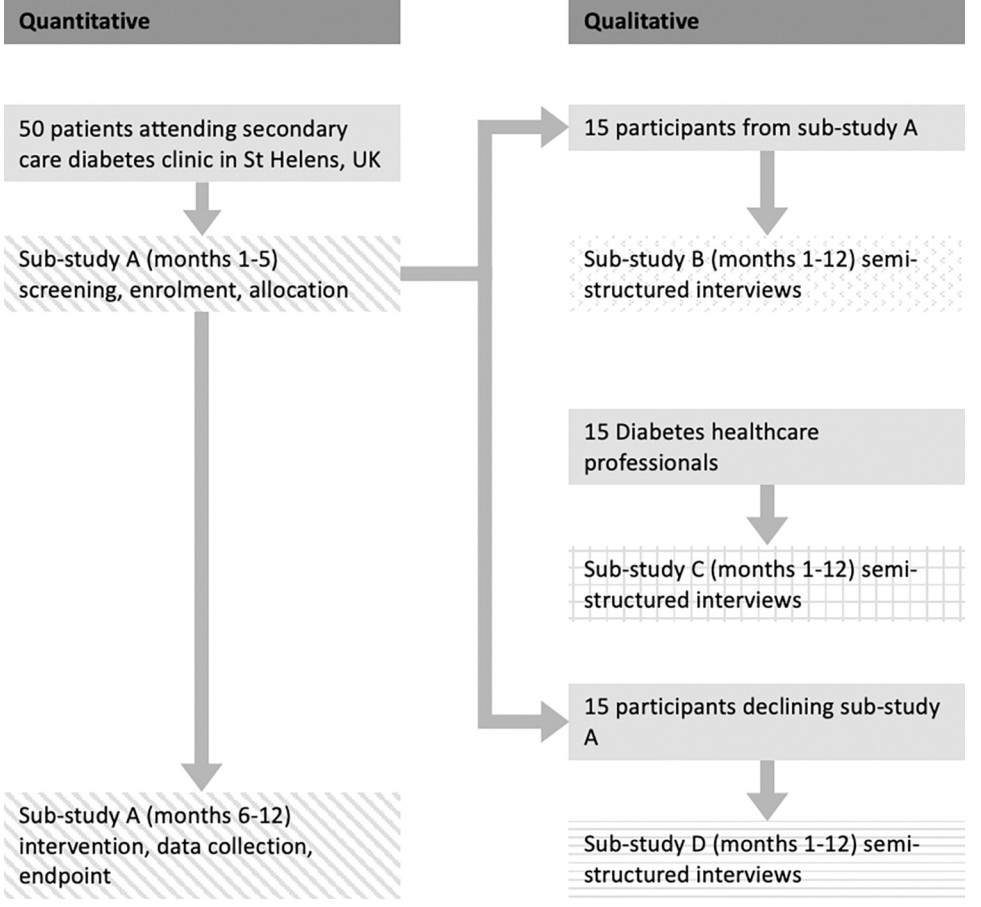

**Fig 2. Flowchart demonstrating the study process and timeline.**

In order to triangulate preliminary quantitative data obtained from sub-study A, a selection of participants (see Allocation) will be recruited for recorded semi-structured telephone interviews (sub-study B). Healthcare professionals involved in delivering secondary diabetes care will be invited via email to take part in a recorded semi-structured interview (sub-study C). Those declining to take part in sub-study A will be offered a survey and recorded semi-structured interview evaluating barriers to participation in research (sub-study D).

## Allocation

Participants enrolled in sub-study A will be randomised using a stratified (strata: type 1 diabetes; type 2 diabetes), random permuted block strategy [38, 39] 1:1 into intervention Group A or Group B, using an online randomisation service [40]. For sub-study B, a stratified (strata: type 1 diabetes; type 2 diabetes), purposive sample of participants in sub-study A will be recruited for recorded semi-structured interviews. For sub-study C, a convenience sample of healthcare professionals directly involved in the care of people with diabetes will be recruited for recorded semi-structured interviews. For sub-study D, a stratified (strata: type 1 diabetes; type 2 diabetes), purposive sample of participants declining enrolment into sub-study A will be recruited for recorded semi-structured interviews and survey.

## Sample size

A formal power calculation was not undertaken as this is usually not required for a feasibility study [41]. Influential papers [42, 43] on the sample size requirements for feasibility studies have suggested between 24 and 50 participants will be required to estimate the standard deviation of an outcome measure (often needed in the calculation of sample sizes in larger studies). To target the upper limit of these suggestions, it is projected that 72 eligible candidates will be required to recruit 50 participants, based on rates of recruitment from similarly structured studies [44, 45] with loss to follow-up expected to be less than 20 percent [46]. Depending on eligibility and recruitment rates, the sample size may have to be adapted.

For qualitative study aspects (sub-studies B, C and D) an estimated 15 participants will be required for each sub-study. Sample adequacy will be determined using the concept of information power [47].

## Outcomes

**Feasibility and acceptability.** Feasibility and acceptability of the trial will be determined according to rates of participant eligibility, recruitment, retention and questionnaire response rates. Further data on acceptability will be captured using an end-of-study survey. In addition, barriers to participation in research will be recorded. The feasibility and acceptability data will identify if progression to a larger study is recommended.

**Preliminary effectiveness.** Examination of the effectiveness of the intervention is exploratory given that the study is not sufficiently powered. Preliminary effectiveness of the intervention on glycated haemoglobin levels and psychometric outcomes (see Measures) will be measured as the key variables of interest with measurements pre-intervention and endpoint (Fig 1).

**Demographics and clinical characteristics.** Baseline data on NHS number, age, sex, ethnicity, diabetes type, diabetes duration, postcode, point-of-care glycated haemoglobin, blood pressure, body mass index and educational status. Follow-up point-of-care glycated haemoglobin, blood pressure and body mass index at three and six months will be recorded. Prescribed glucose-lowering medications and changes to medications will be recorded from medical records at each timepoint.

## Measures

**Feasibility.**  Rates of eligibility, recruitment, retention and questionnaire response will be measured. Barriers to participation and reasons for dropout will be recorded.

**Study acceptability.**  An 8-item Likert-type acceptability survey will evaluate the study processes, participant-facing documents, intervention, understanding of glycated haemoglobin targets, time commitments and psychometric questionnaires. Participants will be asked to rate a statement on each item (1 = 'strongly disagree', 5 = 'strongly agree').

**Health-related quality of life.**  Health-related quality of life will be measured by Euro-QoL's EQ-5D-5L questionnaire [48]. The EQ-5D-5L consists of two pages: the EQ-5D descriptive system and the EQ visual analogue scale (EQ-VAS). The descriptive system measures dimensions of mobility, self-care, usual activities, pain/discomfort and anxiety/depression. Each dimension has five levels: 'no problems', 'slight problems', 'moderate problems', 'severe problems' and 'extreme problems'. Participants indicate their health state ticking the box next to the most appropriate statement in each of the five dimensions. This results in a 1-digit number for each dimension. The numbers are combined into a 5-digit number describing the participant's overall health state (e.g. '11111' being the state of full health). The EQ-VAS records the participants self-rated health visually on a scale from 'The worst health you can imagine' to 'The best health you can imagine'. The EQ-5D-5L tool has good internal consistency, with a Cronbach's alpha between 0.76 and 0.83 reported in diabetes populations [49, 50].

**Diabetes-related distress.**  The Problem Areas in Diabetes questionnaire [51] is a 20-item questionnaire in which each item relates to a different aspect of diabetes-related psychosocial distress (rated 0 = 'not a problem' to 4 = 'serious problem'). The scores of each item are added up and multiplied by 1.25, generating a total score between $0 - 100$. Higher scores indicate a high degree of diabetes-related distress. Participants scoring 40 or higher may be at the level of emotional burnout and warrant special attention. The tool has good internal consistency, with a Cronbach's alpha of 0.9 [52].

**Self-care.**  Self-care in diabetes will be measured by the Summary of Diabetes Self-care Activities questionnaire [53]. The Summary of Diabetes Self-care Activities questionnaire measures the diabetes self-care activities of 'general diet', 'specific diet', 'exercise', 'blood sugar testing' and 'foot care' (scored on 'number of good days per week', 0–7). Items are scored separately due to low levels of inter-item correlation. Higher scores indicate higher levels of self-management for each activity. Additionally, self-reported smoking status ('yes' or 'no') and average number of cigarettes smoked per day are recorded. The tool has moderate internal consistency, with a Cronbach's alpha of 0.62 [54].

**Well-being.**  The 12-item Well-being Questionnaire [55] provides a measure of depressed mood, anxiety and various aspects of positive and negative well-being (items rated 0 = 'not at all' to 3 = 'all the time'). The 12-item Well-being Questionnaire outputs a score of general psychological well-being on a scale of $0 - 36$ with higher scores indicating better general well-being. The tool has good internal consistency, with a Cronbach's alpha of 0.87 [56].

**Diabetes-related psychosocial self-efficacy.**  The Diabetes Empowerment Scale-Long Form [57] measures perceived self-efficacy of participants across 28-items using a Likert-type scale from 0 ('strongly disagree') to 5 ('strongly agree'). Higher scores indicate higher levels of psychosocial self-efficacy in diabetes. The tool has good internal consistency, with a Cronbach's alpha of 0.96 [57].

**Point-of-care glycated haemoglobin.**  Glycated haemoglobin, a blood marker measuring the degree of glycaemia over a period of 8–12 weeks, is measured using capillary blood obtained from participants using a lancet device. Automated point-of-care glycated haemoglobin analysis is completed in five minutes on a DCA Vantage machine which is calibrated and

maintained as per NHS laboratory standards. Glycated haemoglobin levels will be recorded at enrolment (to check eligibility), follow-up (as part of study intervention) and endpoint (to evaluate impact of the intervention).

## Data analysis

Data analysis will aim to describe the feasibility of undertaking a larger trial evaluating the psychometric impact of using glycated haemoglobin targets in adults with type 1 and type 2 diabetes. Analysis of feasibility data from sub-study A will be carried out quantitatively and descriptively using International Business Machines Statistical Package for Social Sciences V.25 statistics software [58]. P-values will not be calculated. Analysis will be presented separately by randomisation group as follows:

- Feasibility data will be presented as numbers and percentages.

- Acceptability data will be expressed as frequencies and percentages.

- For continuous variables, baseline characteristics will be expressed as mean (standard deviation) where normally distributed and median (interquartile range) where distributions are skewed. Categorical variables will be expressed as frequencies and percentages.

- Glycated haemoglobin, blood pressure, body mass index values at 0, 3 and 6 months and EQ-5D-5L index, EQ-VAS, SDSCA, PAID, W-BQ12 and DES-LF questionnaire scores at 3 and 6 months. Missing data from validated questionnaires will be handled as described in the questionnaire user guides.

Qualitative data from semi-structured interviews will be transcribed verbatim, coded to highlight key themes and analysed qualitatively using the Framework Method of thematic analysis [59] in NVivo [60] qualitative data analysis support software. Data will be anonymised during transcription. Pre-defined themes of interest alongside inductively derived themes (themes arising from the data during analysis) will used in coding the qualitative data. Deductive themes, consistent with the study aims, will be derived from the interview guides (S2 File) with new codes arising inductively during transcript analysis to enhance the codebook. Coding strategy for transcript analysis will be determined by discussion amongst the coding team followed by consensus on a final coding strategy. Any disagreements will be resolved with input from a senior reviewer. This approach will allow the team to contrast themes from the patient and healthcare professional perspective.

Interviewing participants taking part in the quantitative study aspects will enable a deeper understanding of the themes of glycated haemoglobin target individualisation, mental health in diabetes and study acceptability and their relationship with health-related quality of life, self-care, diabetes-related distress and well-being. Quantitative and qualitative data will be triangulated to inform on the acceptability of the study processes and patient experiences in relation to glycated haemoglobin target-setting.

## Monitoring

Rates of eligibility, recruitment and retention will be monitored and recorded in a detailed database, including reasons for drop-out. Data will be reported monthly to the research team and sponsor. Adverse and serious adverse events will be recorded alongside these data and reported immediately to the sponsor and Chief Investigator who, together with the research team, determine the severity and association of the event with the study intervention to determine necessary actions.

A green (proceed: there are no concerning issues that threaten the success of a future trial), amber (amend: where there are remediable issues, thereafter, proceed with caution), red (stop: there are intractable issues which cannot be remedied) system will be used to determine progression to a larger trial.

For sub-study A, 50 recruits are required. Each week in the diabetes centre, 13 distinct face-to-face diabetes clinic lists are timetabled, with an average of seven patients per clinic. After accounting for expected non-attenders (2-3/clinic), clinic closures (3-4/week) and clinician unavailability (rare), a lower estimate of 156 patients per month will be available for screening, of which 25 percent ($n = 0.25 \times 156 = 39$) are expected to be eligible. To recruit the required number of participants into sub-study A within the timeframe, successful recruitment of 13 participants per month will be required, meaning recruitment rate is expected to be a minimum of 33 percent ($13 \div 39$). Considering these estimates, conservative progression criteria will be: (1) identification of 39 or more eligible patients per month through screening, (2) recruitment of 13 or more participants per month, (3) monitoring of significant off-protocol deviation; (4) retention of 80 percent or more at follow-up, and; (5) questionnaire response rate of 75 percent of more.

Failure to meet these criteria would result in an amber situation. Remediable adjustments will be identified by the investigators and implemented. Failure to meet progression criteria following remediable adjustments of in the absence of remediable adjustments will prompt review to prevent trial progression.

## Ethics and dissemination

The study has been approved by the Health Research Authority in the United Kingdom (Cornwall and Plymouth Research Ethics Committee, REC reference 21/SW/0043, IRAS ID: 291245, 30th April 2021) and Edge Hill University (Health-related Research Ethics Committee, REC reference ETH2021-0017, 31st March 2021). Internal and external peer review has been undertaken by St Helens and Knowsley Teaching Hospitals NHS Trust as the study sponsor organisation (sponsorship reference: STHK-2021-003). Written, informed consent is obtained from all participants prior to participation in line with UK regulatory requirements.

Study data will be stored on NHS password-protected, encrypted computers or locked filing cabinets on NHS sites. All participant data will be anonymised and assigned a unique study code on database entry. A list of patient identifiers linked to study codes is kept separately in case re-identification is necessary. Biological specimens obtained in the study will not be stored and will be safely disposed of in accordance with the UK's Human Tissue Authority Code of Practice (S1 File).

Any minor or substantial amendments to the study protocol will be reviewed by the Health Research Authority and Edge Hill University Health-related Research Ethics Committee. Such amendments will be communicated to the study sponsor, investigators, trial registry and journals accordingly.

Dissemination plans include publication of study results in scientific, peer-reviewed journals and conferences. Participants will be posted a lay summary of the study findings. Findings also form part of one of the author's (S.J.W.) doctoral thesis. On completion of the study, an anonymised trial dataset and related metadata will be made available via Edge Hill University's online Research Data Repository.

## Discussion

With the rising prevalence of diabetes and a drive by Diabetes UK and the UK government to improve mental health awareness, managing the psychological implications of diabetes is of

increasing importance. Psychological comorbidities in diabetes can significantly impact upon self-care and self-management with subsequent detrimental effects on glycated haemoglobin levels in the long-term. Gaining new insights into the psychological impact that diabetes care processes have on people with diabetes may highlight areas where improvements can be made. This study will output valuable data on the feasibility of studying psychometric outcomes in people with diabetes and preliminarily demonstrate the impact glycated haemoglobin targets have on psychological metrics in people with diabetes. Furthermore, this study will provide qualitative viewpoints of the experiences of participants on study acceptability, the views of participants on the use of glycated haemoglobin targets and ways in which glycated haemoglobin target usage impacts upon their well-being.

Recruitment for quantitative aspects of the study began in June 2021. All participants have been recruited, with follow-up and data collection expected to be completed by April 2022. Qualitative semi-structured interview components of the study will take place during this time period.

## Conclusion

The present mixed-methods study protocol aims to provide a novel insight into the psychological implications of glycated haemoglobin target-setting in people with diabetes in secondary care, alongside testing the feasibility of undertaking a larger project of this nature.

## Supporting information

**S1 Table. SPIRIT (2013) checklist of standard protocol items.**
(DOCX)

**S1 File. Description of the handling of biological specimens.**
(DOCX)

**S2 File. HRA approved protocol version 2.0.5.**
(PDF)

## Acknowledgments

This work forms part of the first author's PhD. S.J.W. is supported by the Department of Diabetes and Endocrinology, St Helens and Knowsley Teaching Hospitals NHS Trust, UK.

## Author Contributions

**Conceptualization:** Samuel J. Westall, Simon Watmough, Ram Prakash Narayanan, Greg Irving, Kevin Hardy.

**Data curation:** Samuel J. Westall.

**Formal analysis:** Samuel J. Westall.

**Funding acquisition:** Samuel J. Westall, Kevin Hardy.

**Investigation:** Samuel J. Westall.

**Methodology:** Samuel J. Westall.

**Project administration:** Samuel J. Westall.

**Software:** Samuel J. Westall.

**Supervision:** Simon Watmough, Ram Prakash Narayanan, Greg Irving, Kevin Hardy.

**Validation:** Greg Irving.

**Visualization:** Samuel J. Westall.

**Writing – original draft:** Samuel J. Westall.

**Writing – review & editing:** Samuel J. Westall, Simon Watmough, Ram Prakash Narayanan, Greg Irving, Kevin Hardy.

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
