## [Decision Letter · Decision Letter 0]

1 May 2022

PONE-D-21-33122Psychometric and biomedical outcomes of glycated haemoglobin target-setting in adults with type 1 and type 2 diabetes: Protocol for a mixed-methods parallel-group randomised feasibility studyPLOS ONE

Dear Dr. Westall,

Thank you for submitting your manuscript to PLOS ONE. After careful consideration, we feel that it has merit but does not fully meet PLOS ONE’s publication criteria as it currently stands. Therefore, we invite you to submit a revised version of the manuscript that addresses the points raised during the review process.

We look forward to receiving your revised manuscript.

Kind regards,

Jianhong Zhou

Staff Editor

PLOS ONE

Journal Requirements:

3. Thank you for stating the following in the Acknowledgments Section of your manuscript: "The authors thank Mrs Jeanette Anders for assistance with ethical approval and sponsorship of the research, Dr Niall Furlong for advice and support and Dr Sumudu Bujawansa for pastoral support."

Please remove any funding-related text from the manuscript and let us know how you would like to update your Funding Statement. Currently, your Funding Statement reads as follows: "The corresponding author (S.J.W.) is a PhD student funded by the Department of Diabetes and Endocrinology, St Helens and Knowsley Teaching Hospitals NHS Trust (STHK). The study is sponsored by STHK (sponsorship reference STHK-2021-003). STHK are not involved in the study design, the preparation of this paper or the decision to submit the paper for publication. The funding body will not be involved in data collection, analysis or interpretation. St Helens and Knowsley Teaching Hospitals NHS Trust (STHK): https://www.sthk.nhs.uk"

5. Please note that in order to use the direct billing option the corresponding author must be affiliated with the chosen institute. Please either amend your manuscript to change the affiliation or corresponding author, or email us at plosone@plos.org with a request to remove this option.

Reviewers' comments:

Reviewer's Responses to Questions

**Comments to the Author**

1. Does the manuscript provide a valid rationale for the proposed study, with clearly identified and justified research questions?

Reviewer #1: Yes

2. Is the protocol technically sound and planned in a manner that will lead to a meaningful outcome and allow testing the stated hypotheses?

Reviewer #1: Yes

3. Is the methodology feasible and described in sufficient detail to allow the work to be replicable?

Reviewer #1: Yes

4. Have the authors described where all data underlying the findings will be made available when the study is complete?

Reviewer #1: Yes

5. Is the manuscript presented in an intelligible fashion and written in standard English?

Reviewer #1: Yes

6. Review Comments to the Author

You may also provide optional suggestions and comments to authors that they might find helpful in planning their study.

Reviewer #1: This protocol paper was a pleasure to read and is exploring a very important yet under researched topic of how clinician-initiated HbA1c testing and setting A1c targets affects the wellbeing and self-care of people living with diabetes. The manuscript was easy to read and the introduction had a strong rationale leading to a good description of the methods and analyses. The mixed-methods design is appropriate. My only concern was the statistical analysis section. As this is a feasibility study and underpowered the authors should not be conducting any statistical tests (e.g. t-tests) looking for differences from baseline to post-intervention but rather simply reporting means and SDs. Also the qualitative analysis section was very brief.

7. PLOS authors have the option to publish the peer review history of their article (what does this mean?). If published, this will include your full peer review and any attached files.

Reviewer #1: No

---

## [Author Response · Author response to Decision Letter 0]

14 May 2022

Reviewer comments and author responses:

Reviewer #1: This protocol paper was a pleasure to read and is exploring a very important yet under researched topic of how clinician-initiated HbA1c testing and setting A1c targets affects the wellbeing and self-care of people living with diabetes. The manuscript was easy to read and the introduction had a strong rationale leading to a good description of the methods and analyses. The mixed-methods design is appropriate. My only concern was the statistical analysis section. As this is a feasibility study and underpowered the authors should not be conducting any statistical tests (e.g. t-tests) looking for differences from baseline to post-intervention but rather simply reporting means and SDs. Also the qualitative analysis section was very brief.

Author response:

Thank you. We agree with the reviewer’s comments regarding statistical analysis and have adjusted the manuscript accordingly. We have also expanded the qualitative analysis section with additional detail. (Pages 15 and 16, lines 328–354)

Kind Regards, 

Sam Westall (corresponding author)

---

## [Decision Letter · Decision Letter 1]

25 Jul 2022

PONE-D-21-33122R1Psychometric and biomedical outcomes of glycated haemoglobin target-setting in adults with type 1 and type 2 diabetes: Protocol for a mixed-methods parallel-group randomised feasibility studyPLOS ONE

Dear Dr. Westall,

Thank you for submitting your manuscript to PLOS ONE. After careful consideration, we feel that it has merit but does not fully meet PLOS ONE’s publication criteria as it currently stands. Therefore, we invite you to submit a revised version of the manuscript that addresses the points raised during the review process.

We look forward to receiving your revised manuscript.

Kind regards,

Jamie Royle

Staff Editor

PLOS ONE

Journal Requirements:

Reviewers' comments:

Reviewer's Responses to Questions

**Comments to the Author**

1. Does the manuscript provide a valid rationale for the proposed study, with clearly identified and justified research questions?

Reviewer #2: Yes

2. Is the protocol technically sound and planned in a manner that will lead to a meaningful outcome and allow testing the stated hypotheses?

Reviewer #2: Yes

3. Is the methodology feasible and described in sufficient detail to allow the work to be replicable?

Reviewer #2: Yes

4. Have the authors described where all data underlying the findings will be made available when the study is complete?

Reviewer #2: Yes

5. Is the manuscript presented in an intelligible fashion and written in standard English?

Reviewer #2: Yes

6. Review Comments to the Author

You may also provide optional suggestions and comments to authors that they might find helpful in planning their study.

Reviewer #2: I will focus on methods and reporting, as the statistical reviewer. Methods and approach to the sample are appropriate. I only have a couple of minor points to raise.

1) eligibility criteria. There are numerous ongoing interventions and the participants need to be at least asked about such participation so that can be included in the analyses (rather than as an additional exclusion criterion).

2) In the UK, under the Quality and outcomes framework, HbA1c targets are incentivised at the practice level. Does that run a risk of contaminating the trial in any way? Treatment information will be collected the authors say, but needs to be collected very well and possibly not be self-reported. Detailed information is needed in this space.

7. PLOS authors have the option to publish the peer review history of their article (what does this mean?). If published, this will include your full peer review and any attached files.

Reviewer #2: No

---

## [Author Response · Author response to Decision Letter 1]

19 Aug 2022

14th August 2022

Re: PLOS ONE Decision: Revision required [PONE-D-21-33122] - [EMID:c1e2bb034c361eaf]

Dear Editor and Reviewers, 

On behalf of my co-authors, I would like to thank you for reviewing our article entitled “Psychometric and biomedical outcomes of glycated haemoglobin target-setting in adults with type 1 and type 2 diabetes: Protocol for a mixed-methods parallel-group randomised feasibility study” with opportunity to revise and resubmit. 

We found the reviewer’s comments helpful in revising the manuscript and have carefully considered their suggestions. We revised the manuscript in accordance with the reviewer’s comments. 

Below, we have outlined each of the reviewer comments followed by our responses, prefaced “Author response”. Corresponding changes are highlighted in the tracked manuscript text file in red using the ‘track changes’ function. 

Yours Sincerely, 

Review comments and author responses

Reviewers' comments:

Reviewer's Responses to Questions 

Comments to the Author

1. Does the manuscript provide a valid rationale for the proposed study, with clearly identified and justified research questions?

Reviewer #2: Yes

2. Is the protocol technically sound and planned in a manner that will lead to a meaningful outcome and allow testing the stated hypotheses?

Reviewer #2: Yes

3. Is the methodology feasible and described in sufficient detail to allow the work to be replicable?

Reviewer #2: Yes

4. Have the authors described where all data underlying the findings will be made available when the study is complete?

Reviewer #2: Yes

5. Is the manuscript presented in an intelligible fashion and written in standard English?

Reviewer #2: Yes

6. Review Comments to the Author

You may also provide optional suggestions and comments to authors that they might find helpful in planning their study.

Reviewer #2: I will focus on methods and reporting, as the statistical reviewer. Methods and approach to the sample are appropriate. I only have a couple of minor points to raise.

1) eligibility criteria. There are numerous ongoing interventions and the participants need to be at least asked about such participation so that can be included in the analyses (rather than as an additional exclusion criterion).

2) In the UK, under the Quality and outcomes framework, HbA1c targets are incentivised at the practice level. Does that run a risk of contaminating the trial in any way? Treatment information will be collected the authors say, but needs to be collected very well and possibly not be self-reported. Detailed information is needed in this space.

Author response:

6 – 1) Thank you for your comments. 

We have added a sentence under the eligibility sub-heading to clarify that eligible individuals will be invited to take part and that, where declined, reasons for decline will be recorded (lines 148, 180, 183).

6 – 2) HbA1c target incentivisation was considered as a potential confounder in this trial, however with recruitment occurring from a secondary care diabetes clinic, primary care target incentivisation during the study period is not anticipated to influence study results. Certainly, if recruitment was also being undertaken from primary care, we would need to account for this. We have updated the manuscript for clarification (line 160).

The authors agree that the collection of treatment information must be thorough and accurate. These data will be obtained from medical records rather than relying on self-reported data. We have updated the manuscript to clarify this (line 242).

7. PLOS authors have the option to publish the peer review history of their article (what does this mean?). If published, this will include your full peer review and any attached files.

Do you want your identity to be public for this peer review? For information about this choice, including consent withdrawal, please see our Privacy Policy.

Reviewer #2: No

------END OF RESPONSE------

---

## [Decision Letter · Decision Letter 2]

27 Sep 2022

Psychometric and biomedical outcomes of glycated haemoglobin target-setting in adults with type 1 and type 2 diabetes: Protocol for a mixed-methods parallel-group randomised feasibility study

PONE-D-21-33122R2

Dear Dr. Westall,

We’re pleased to inform you that your manuscript has been judged scientifically suitable for publication and will be formally accepted for publication once it meets all outstanding technical requirements.

Kind regards,

Walid Kamal Abdelbasset, Ph.D.

Academic Editor

PLOS ONE

Additional Editor Comments (optional):

Reviewers' comments:

Reviewer's Responses to Questions

**Comments to the Author**

1. Does the manuscript provide a valid rationale for the proposed study, with clearly identified and justified research questions?

Reviewer #2: Yes

Reviewer #3: Yes

2. Is the protocol technically sound and planned in a manner that will lead to a meaningful outcome and allow testing the stated hypotheses?

Reviewer #2: Yes

Reviewer #3: Yes

3. Is the methodology feasible and described in sufficient detail to allow the work to be replicable?

Reviewer #2: Yes

Reviewer #3: Yes

4. Have the authors described where all data underlying the findings will be made available when the study is complete?

Reviewer #2: Yes

Reviewer #3: Yes

5. Is the manuscript presented in an intelligible fashion and written in standard English?

Reviewer #2: Yes

Reviewer #3: Yes

6. Review Comments to the Author

You may also provide optional suggestions and comments to authors that they might find helpful in planning their study.

Reviewer #2: I am satisfied with the authors' responses and I do not have anything else to add.........................

Reviewer #3: Accoing to the previous review and from my point of view the authors had adressed all the required modifications

7. PLOS authors have the option to publish the peer review history of their article (what does this mean?). If published, this will include your full peer review and any attached files.

Reviewer #2: No

Reviewer #3: No

---

## [Editor Report · Acceptance letter]

20 Oct 2022

PONE-D-21-33122R2 

Psychometric and biomedical outcomes of glycated haemoglobin target-setting in adults with type 1 and type 2 diabetes: Protocol for a mixed-methods parallel-group randomised feasibility study 

Dear Dr. Westall:

I'm pleased to inform you that your manuscript has been deemed suitable for publication in PLOS ONE. Congratulations! Your manuscript is now with our production department. 

Kind regards, 

on behalf of

Dr. Walid Kamal Abdelbasset 

Academic Editor

PLOS ONE